# ECG Enhancement and R-Peak Detection Based on Window Variability

**DOI:** 10.3390/healthcare9020227

**Published:** 2021-02-18

**Authors:** Lu Wu, Xiaoyun Xie, Yinglong Wang

**Affiliations:** 1College of Computer Science and Engineering, Shandong University of Science and Technology, Qingdao 266590, China; lwu@qlu.edu.cn; 2Shandong Artificial Intelligence Institute, Qilu University of Technology (Shandong Academy of Sciences), Jinan 250014, China; xiexy@sdas.org

**Keywords:** ECG, enhancement, R-peaks, squared window variance transform (SWVT), adaptive thresholds

## Abstract

In ECG applications, the correct recognition of R-peaks is extremely important for detecting abnormalities, such as arrhythmia and ventricular hypertrophy. In this work, a novel ECG enhancement and R-peak detection method based on window variability is presented, and abbreviated as SQRS. Firstly, the ECG signal corrupted by various high or low-frequency noises is denoised by moving-average filtering. Secondly, the window variance transform technique is used to enhance the QRS complex and suppress the other components in the ECG, such as P/T waves and noise. Finally, the signal, converted by window variance transform, is applied to generate the R-peaks candidates, and the decision rules, including amplitude and kurtosis adaptive thresholds, are applied to determine the R-peaks. A special squared window variance transform (SWVT) is proposed to measure the signal variability in a certain time window, and this technique reduces false detection rate caused by the various types of interference presented in ECG signals. For the MIT-BIH arrhythmia database, the sensitivity of R-peak detection can reach 99.6% using the proposed method.

## 1. Introduction

The electrocardiogram (ECG) is a random and unstable signal that records the electrical activity of the heart, and the ECG signals obtained in different environments and from individuals are significantly different. The P wave, QRS complex and T wave are the main components in the ECG waveform (see Figure 1) [1], and the accurate detection of them is important to ECG signal analysis. The QRS complex is the dominant feature of the ECG signal and its accurate detection is an important issue in many clinical conditions [2]; for example, the RR interval is used for heart rate measurement and diagnosis of several abnormalities—ventricular hypertrophy [3], conduction abnormalities [4], etc.

The detection of the QRS complex is the first step in all kinds of automatic feature extractions for ECG signals [5]. Unluckily, there are great challenges for automated detection because the morphologies and amplitudes of many normal QRS complexes are like the abnormal QRS complexes. The superimposed noise in the ECG signal makes this problem more severe. Furthermore, the P/T waves with higher amplitude can interfere with the detection of the QRS complex. Therefore, the first step of R-peak detection is signal denoising, and then the QRS complexes are enhanced and detected.

Many denoising methods [6,7,8,9,10,11,12,13] for ECG signals have been proposed, including digital filtering [14], morphological filtering [15,16] and decomposition-based denoising methods [13,17,18]. Among them, the digital filters are widely used to remove the noise of a certain frequency range because of their simplicity and efficiency; morphological filters [16] are often used to smooth the ECG signal, but the size and shape is hard to determine; decomposition-based denoising methods such as wavelet transform [16] and empirical mode decomposition (EMD) [17] methods decompose the signal into a series of modes and set some modes to 0 to remove noise. Thus, most of the noises can be well addressed by the methods mentioned above.

For QRS detection, most QRS detection methods [19,20,21,22,23] are researched based on the three steps—denoising, QRS complex enhancement and decision rule creation. The stages of QRS complex enhancement and decision rule creation are different and result better or worse results. In [24], the Pan-Tompkins algorithm was proposed for R-peak detection, which includes five steps, namely, band-pass filtering, derivation, squaring, moving-window integration and adjustment of thresholds. In [25], the wavelet transforms (WT) for detecting QRS complexes is proposed, in which the multiscale feature of WT is used to distinguish QRS complexs from P/T waves or noise. In [26], an EMD-based method for QRS detection was proposed; unlike the wavelet methods, the EMD methods have overcome their mode-mixing problem, having emerged as powerful time-frequency decomposition tools. However, the preprocesses of the above methods need to be carefully designed, and multiple nonlinear transform techniques are necessary to enhance the QRS complex.

In the proposed method, the noisy signal is first denoised by moving-average filtering, and it can be easily replaced by other denoising methods. In the stage of ECG enhancement, multiple methods were usually employed at the same time for the most studies; however, the proposed window variance transform technique is very effective for highlighting R-peaks and suppressing P/T waves and noise in the ECG signal, and all R-peaks candidates can be well identified. Even if additional techniques are added, the candidates will not be greatly improved. Hence, it is sufficient to use only window variance transform for signal enhancement. Furthermore, two adaptive thresholds related to amplitude and kurtosis are computed for recognizing the locations of R-waves. The experimental results demonstrate that the proposed QRS detection method can accurately locate the R-wave positions. The main contributions of this paper are as follows: (1) A novel QRS detection method is proposed to accurately locate the positions of the R-waves. (2) An efficient QRS-enhancement technique is designed.

This paper is organized as follows. In Section 2, we give a brief introduction of the proposed method. In Section 3, we show and discuss the experimental results of the proposed algorithm, in which the evaluation of algorithm is accomplished with the ECG data from the MIT/BIH arrhythmia database. Concluding remarks are given in Section 4.

## 2. Materials and Methods

### 2.1. Data

The MIT-BIH database [27] contains 48 ECG records; each record contains 30 min and was sampled at 360 Hz. There are two leads in each record and the first lead was used in the experiment. Locations of R-peaks have been annotated by two or more cardiologists independently for all records.

### 2.2. Overview of the Algorithms

The proposed method for R-peak detection is divided into three stages, including noise removal, ECG enhancement and decision rules. Those stages are illustrated in Figure 2. In the proposed method, the ECG enhancement stage plays the most critical role; it highlights the QRS complex and suppresses the other components in the ECG signal. A brief description of each stage in Figure 2 is presented in the following subsections.

#### 2.2.1. Noise Removal

Since the signal is often corrupted by various noise—such as the high-frequency noise, baseline wander noise and artifacts—in this section, to maintain the original morphological features of the ECG signal, the moving average filtering is applied to smooth noise, and the formula of moving average filters can be described as follows:(1)$$y_{n}^{\prime }=\frac{1}{M}\sum_{m=1}^{M}\left(y_{n-m}\right)$$
where $y_{i}$ and $y_{i}^{\prime }$ represent the amplitudes of the *i*th sample in the original ECG signal and denoised signal, respectively; *M* represents the filter length, which is set to 5 samples.

#### 2.2.2. ECG Enhancement

In this section, a novel ECG enhancement technique called window variance transform is proposed. The variance of an signal segment is used to highlight all R-peaks and suppress the other components. The proposed squared window variance transform (SWVT) is summarized as follows:

1.Non-overlapping windowLocal peaks $P=[p_{1},p_{2},\ldots ,p_{N}]$ are first detected using the denoised signal. Then, *P* is applied to generate an non-overlapping window N, $W_{P}=\left[w_{p_{1}},w_{p_{2}},\ldots ,w_{p_{N}}\right]$, and the *i*th window is determined by using $w_{p_{i}}=\left[p_{i}-w/2,p_{i}+w/2\right]$, where *w* is the window size. To avoid including multiple waveforms in one window, the minimum width of wave T is used as a reference to set the range of w from 40 to 60 ms. Next, for the non-peak samples between two peaks, they are divided into M non-overlapping windows: $W_{NP}=\left[w_{np_{1}},w_{np_{2}},\ldots ,w_{np_{M}}\right]$; the window size is 2∗w. Finally, the $W_{P}$ and $W_{NP}$ are merged to generate the N+M non-overlapping windows $W=[w_{1},w_{2},\ldots ,w_{N+M}]$.2.Window variance transformTo detect the R-peaks, the window variance transform is used to enhance the QRS complexes; that is, the denoised signal is transformed to the $WVT=[v_{i},v_{2},\ldots ,v_{N+M}]$, in which
(2)$$v_{j}=\frac{1}{l_{j}}\sum_{t=0}^{l_{j}}{\left(y_{t}^{\prime }-\mu_{j}\right)}^{2}$$
where j=[1,2,…,N+M] are the indexes of windows; $l_{j}$ is the number of samples in *j*th window; $\mu_{j}$ is the amplitude mean of the *j*th window samples.3.SquaringTo further enhance the QRS complexes, the squaring operator is applied in the WVT, that is, the squared window variance transform (SWVT) is described as follows:
(3)$$SWVT=[v_{1}^{2},v_{2}^{2},\ldots ,v_{N+M}^{2}]$$

As shown in Figure 3 and Figure 4, the result of SWVT significantly suppressed the effect of the P/T wave and enhanced the QRS complex. From Figure 4, it can be seen that SWVT is also beneficial for the an ECG signal with inverse R peaks. Thus, the proposed QRS complex enhancement technique is reasonable for R-peak detection.

#### 2.2.3. R-Peak Detection

Due to R-peaks being different from other waveforms in terms of morphology and amplitude, most existing detection methods use amplitude thresholds for R-peak detection, which is less robust to noisy signals. However, QRS complexes usually have significant differences in amplitude and kurtosis compared with those of the other waveforms in ECG signals; thus, the amplitude and kurtosis thresholds are applied for R-peak detection. The detailed achievement is described as follows:
1.Generation of the R-peaks candidate set

The transformed signal SWVT not only highlights the R-waves but also suppresses the P/T waves, and the screening of the R-peaks candidate set can be completed by setting the amplitude threshold. Considering that the R-peaks which are disturbed by noise may have lower amplitudes during the transformation, the 90th percentile of SWVT and the mean of SWVT are applied to balance the amplitude threshold $T_{c}$; that is,
(4)$$T_{c}=0.5*\left(0.75*per_{\left(SWVT,90\right)}+0.25*\mu_{SWVT}\right)$$
where $per_{\left(SWVT,90\right)}$ and $\mu_{SWVT}=\frac{1}{N+M}\sum_{t=0}^{N+M}{SWVT}_{t}$ are the 90th percentile value and the mean value of SWVT, respectively; $T_{c}$ denotes the positive threshold determining the transformed amplitude of candidate selection.

Then the amplitude threshold $T_{c}$ is used for the extraction of R-peak candidates set $S_{c}$:
(5)$$S_{c}=\left\{w_{j}|v_{j}>T_{c},\;j=1,2,3,\ldots ,N+M\right\}$$

2.Generation of the R-peaks with decision rules

(1) Adjusting the RR interval: the RR interval is another important feature of R-peaks. In this study, the RR interval is determined by two RR interval averages. One is the average of the eight most-recent beats $RR_{recent}$; the other is the average of the detected R-peaks, $RR_{all}$—that is,
(6)$$RR=0.75*RR_{recent}+0.25*RR_{all}$$

For the first RR interval, the lower limit of the RR interval and the higher limit of the RR interval are set to 200 and 360 ms, respectively. However, for irregular heart rates, if an R peak is not found during the interval specified by the 1.66∗RR, a search-back technique is necessary to look back in time for the R peak.

(2) Adjusting the thresholds: To better detect the R-peaks, the amplitude and kurtosis thresholds $T_{a}$ and $T_{k}$ are automatically adjusted to float over R-peak candidate points and detected R-peaks.
(7)$$\begin{array}{c} T_{a}=0.75*\left(\alpha *PER_{A_{c}}+\left(1-\alpha \right)*MEAN_{A_{R}}\right) \end{array}$$
(8)$$\begin{array}{c} T_{k}=0.75*\left(\alpha *PER_{K_{c}}+\left(1-\alpha \right)*MEAN_{K_{R}}\right) \end{array}$$
where $PER_{A_{c}}$ and $PER_{K_{c}}$ are the 90th percentile values of the amplitude and kurtosis in the candidate set $S_{c}$, respectively. Similarly, $MEAN_{A_{R}}$ and $MEAN_{K_{R}}$ are mean values of the amplitude and kurtosis in the peak set $S_{R}$, respectively. α is a hyperparameter, and it is used to control the weight of R-wave amplitude and kurtosis for adjusting thresholds.

For irregular heart rates, the lower $T_{a}$ and $T_{k}$ are needed for back searching for missing R-peaks; hence, two thresholds are reduced by half to avoid missing beats:
(9)$$\begin{array}{c} T_{a}\leftarrow 0.5*T_{a} \end{array}$$
(10)$$\begin{array}{c} T_{k}\leftarrow 0.5*T_{k} \end{array}$$

(3) Generation of the R-peaks: according to the thresholds aforementioned, we designed some decision rules to generate the R peaks $S_{R}$. For each of the peak in $S_{c}$, if an RR interval is between 0.92∗RR and 1.16∗RR, a judgment is made to determine whether the current peak is the R peak; the decision rules can be described as follows:
(11)$$\begin{array}{c} S_{R}=\left\{S_{C_{l}}|y_{C_{l}}^{\prime }>T_{a}\;\mathrm{and}\;k_{C_{l}}>T_{k}\;l=1,2,\ldots ,L\right\} \end{array}$$
(12)$$\begin{array}{c} 0.92*RR<S_{C_{l}}-S_{R_{last}}<1.16*RR \end{array}$$
where $C_{l}$ and $y_{C_{l}}^{\prime }$ represent the $l_{th}$ candidate peak and the amplitude of the $l_{th}$ candidate peak; $k_{C_{l}}$ is the kurtosis of the $l_{th}$ candidate peak; $S_{R_{last}}$ refers to the position of the last R-peak in the current R-peaks set.

## 3. Results

### 3.1. Metrics for Performance Evaluation

In this study, three metrics were used to evaluate the performance of R-peak detection in the simulation experiment, sensitivity (SEN), positive predicative value (PPV) and cumulative statistical index (CSI) [28], which are described as follows.
(13)$$\begin{array}{c} \mathrm{SEN}=\frac{\mathrm{TP}}{\mathrm{TP}+\mathrm{FN}}*100 \end{array}$$
(14)$$\begin{array}{c} \mathrm{PPV}=\frac{\mathrm{TP}}{\mathrm{TP}+\mathrm{FP}}*100 \end{array}$$
(15)$$\begin{array}{c} \mathrm{CSI}=\frac{1}{2}\left(\mathrm{SEN}+\mathrm{PPV}-\mathrm{FPR}-\mathrm{FNR}\right)*100 \end{array}$$
(16)$$\begin{array}{c} F_{d}=\frac{\mathrm{FP}+\mathrm{FN}}{\mathrm{TP}+\mathrm{FN}+\mathrm{FP}} \end{array}$$
where TP,FN and FP represent true positive (correctly identified R peak), false negative (missing R peak) and false positive (wrongly identified R peak), respectively. FPR is the false positive rate FP/(FP+TP) and FNR is the false negative rate FN/(FN+TP). $F_{d}$ was employed to measure the detection error rate. Larger SEN, PPV and CSI values mean better detection performance; meanwhile, a lower $F_{d}$ is expected.

### 3.2. Experimental Data and Environment

In the experiment, 30 ECG records from the MIT-BIH arrhythmia database were adopted to evaluate the effectiveness of the proposed R-peak detection method. The length of all signals was 30 min. The classic PT [24], XQRS [29] and RSlope [30] methods are introduced for comparison. These methods were implemented with Python and Matlab, and the parameters were set according to the original papers. All experimental results were performed on Linux x86_64, Matlab: version R2018b and Python: version 3.7.6.

### 3.3. Detailed Accuracy of the MIT-BIH Database

In the first experiment, the R-peak detection results were derived with the proposed method when applied to all 30 ECG signals from MIT-BIH dataset. In Table 1, the TP, FP, FN, SEN, PPV and CSI of each record are listed. From Table 1, we find that both SEN and PPV still show very high accuracy for most of the records. The proposed method did not perform well in several records (114, 207, 208, 214, 228, 232), for which the SEN or PPV were lower than 99%. In addition, the CSI values for most records were lower than 98%, because some records showed higher FP and FN.

### 3.4. Accuracy Evaluation and Comparison

In the second experiment, the overall performance of the proposed method on MIT-BIH database is shown in Table 2, which also contains three comparison results. In this paper, the detected locations of R-peaks are very close to the reference locations, and some R-peaks were shifted by less than 75 ms compared with the reference R peaks. It can be seen from Table 2 that 99.03% of total beats can be detected correctly by the proposed method; SEN and PPV were 99.65% and 99.39%, respectively. Moreover, the CSI of the proposed method was higher than 99%—better than those of the methods used for comparison.

Table 3 was drawn based on Table 2, which shows the FP and FN comparison results among the four methods. In Table 3, the proposed method shows the least detection errors among the four methods. It is worth noticing that there is a discrepancy between the accuracy reported here and those reported in other methods [5,31]. This is possibly because the different tolerances for counting a detection as a TP are set. In this study, the tolerance is set to ±75 ms of the annotation. In a total of 88,699 beats, there were 482 and 335 beats detected as FP and FN, respectively. Furthermore, the proposed method achieved the minimum detection error of 0.92%. As the proposed method effectively suppressed the P/T wave and noise in the stage of R-peak enhancement, the proposed method significantly reduced the number of FN. Therefore, it is seen that the proposed method has better detection than the other methods.

### 3.5. Visual Display of R-Peak Detection Results

Figure 5 presents an ECG segment with an obvious waveform distortion caused by large baseline wander, and the R-peak detection results produced by different methods. Compared with the reference labels, there are several obvious errors for PT, XQRS and Rslope methods, which are marked by red rectangles and Xs. However, our method can deal with this case very well. In other words, the proposed method effectively suppressed the noise. Then, R-peak detection results were more carefully analyzed using record 208. Figure 6 shows the original ECG signal from 15 to 30 s and the R-peaks locations detected by different detectors. The proposed method can accurately detect the locations of all R-peaks; however, the XQRS and R slope showed some missing R-peaks. This phenomenon may be caused by irregular beats. Furthermore, XQRS demonstrates the worst performance for the wide ectopic beats.

## 4. Discussion

In this study, the performance of the proposed R-peak detection method was tested on MIT-BIH database. Based on the results presented in Table 2, all these methods have excellent R-peak detection capacity with sensitivity rates higher than 99%. That is, most existing methods for R-peak detection have shown excellent performance. However, the three stages for R-peak detection, including preprocessing, enhancement and decision rules, are very different. This paper provided a novel QRS-enhancement R-peak detection method. Next, we analyzed the impacts of key steps and parameters on the experimental results.

### 4.1. ECG Enhancement

As shown in Table 4, the proposed method is easy to perform and does not need complex mathematical calculations compared with the other methods, especially for the stage of enhancement of QRS complexes. For instance, the proposed work only needs one to perform a simple window variance transform to achieve QRS complex enhancement and obtain better detection performance.

### 4.2. The Effect of Using a Kurtosis Threshold

Figure 7 illustrates the distribution comparison between the QRS complexes and other ECG waveforms, in which QRS, P and T waves were extracted from the QT Database [32], and the number of each wave is 100. In Figure 7, the amplitude and kurtosis distributions of the main waveforms in ECG signal are plotted; it can be seen that there is are significant differences between the QRS complex and P/T wave in amplitude and kurtosis values. In statistics, a higher kurtosis means that the increase in variance is caused by extreme differences in low frequencies that are greater or less than the average.

Next, we analyzed the impacts of key steps and parameters on the experimental results. It is well known that most of R-peak detection methods only depend on amplitude thresholds to identify R-peaks, and some mistakes may occur for unobvious R-waves shape. Thus, in this study, two statistical thresholds were used for R-peak identification. In Table 5, we list the performance of using kurtosis threshold on detection results. As shown in Table 5, compared with the performance without using a kurtosis threshold, the proposed method can get fewer FN and more TP, and the improvements to SEN and CSI are 1.21% and 1.08%, respectively. Hence, the kurtosis threshold plays a vital role in R-peak identification, and there is a significant decrease in FN because R-peak features were deeply extracted. As a result, higher CSI and SEN were obtained.

Some of the signals are affected by stretches of noise, baseline wander and artifacts. For example, 222 has some non-QRS waves with highly unusual morphologies (may lead to false positives); 208 has an irregular RR interval (may lead to missing R-peaks). In Figure 5 and Figure 6, the R-peak detection results are shown; the proposed method can achieve annotations the same as the references. However, for the comparison methods, obvious FP and FN occurred. This was due to improper threshold setting or improper noise processing during QRS complex enhancement. However, the proposed method can deal with this case by introducing the simple window variance transform and decision rules. That is, the two thresholds, including amplitude and kurtosis, can distinguish between R-peaks and non-R peaks which are similar to the R-peaks in morphology and amplitude.

### 4.3. Selection of the Weight

In this work, the performance of the proposed method was influenced by the amplitude and kurtosis thresholds; that is, the superparameter α in Equations (Equation 7) and (Equation 8) had a large effect on experimental results. The α is a superparameter which represents the weight of R-peaks amplitude or kurtosis. In Figure 8, we plot the R-peak detection performance of the proposed method with different α values. In Equations (Equation 7) and (Equation 8), larger α means lower thresholds and more FP, which results in lower PPV. On the contrary, smaller α means higher thresholds and can lead to more FN and lower SEN. From Equation 8, we can see that α=0.5 achieved the best CSI; meanwhile, better SEN and PPV were obtained.

However, there are still some limitations to the present study. A signal with unusual morphologies may cause bad performance by the proposed method. The reason is that the R-peak detection result is more relative to the approativate adaptive thresholds, but the amplitude and kurtosis values of the unusual R-peaks are irregular. In addition, the adaptive thresholds depend on the window size, and the selection of window size is another challenge. However, the fixed window may not include all samples in a wave with severe waveform variations, and the results of R-peaks based on window variance transform may not be satisfying. Those limitations have to be solved for future research. Furthermore, we will also consider using other non-curated datasets, such as the NST dataset, and ambulatory ECG dataset obtained by wearable devices, to observe and analyze the preprocessing performance of the proposed method for noisy signals.

## 5. Conclusions

In this paper, a simple and reliable ECG enhancement and R-peaks detection method was proposed, which is easy to perform and does not need complex mathematical calculations. The proposed method for R-peak determination is very robust for signals with larger amplitudes or P/T waves. The sensitivity of detecting R-peaks in the MIT-BIH database using the proposed method can reach 99.65%, which is much better than the most of the existing R-peak detection methods. The experimental results and analysis demonstrate that the proposed R-wave enhancement technique and the decision rules, including amplitude and kurtosis factors, are useful for the recognition of R-peaks. It can be conclude that the proposed method is suitable and reliable for ECG R-peak detection.

## Figures and Tables

**Figure 1 healthcare-09-00227-f001:**
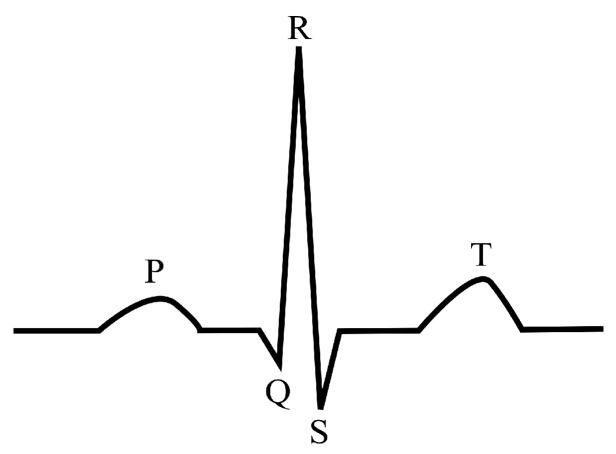
The main waveform in an ECG signal. P wave—Indicates atrial depolarization, or contraction of the atrium; QRS complex—Indicates ventricular depolarization, or contraction of the ventricles; T wave—Indicates ventricular repolarization.

**Figure 2 healthcare-09-00227-f002:**
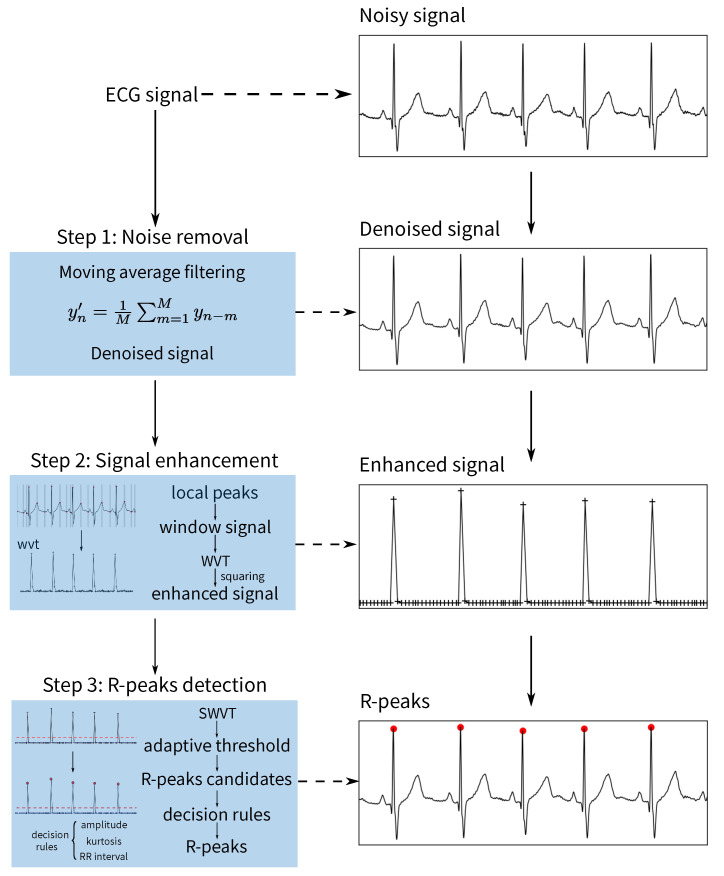
Flowchart of the proposed method. WVT—window variance transform signal.

**Figure 3 healthcare-09-00227-f003:**
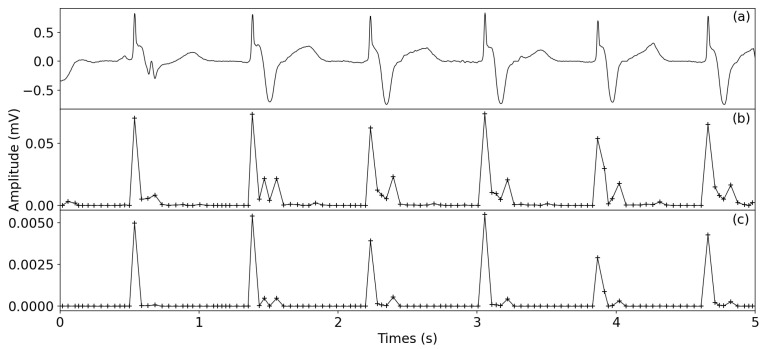
Comparison of original signal and the transformed signals, (**a**) Original signal of 104 from MIT-BIH database. (**b**) WVT—window variance transform signal. (**c**) SWVT—squared window variance transform signal.

**Figure 4 healthcare-09-00227-f004:**
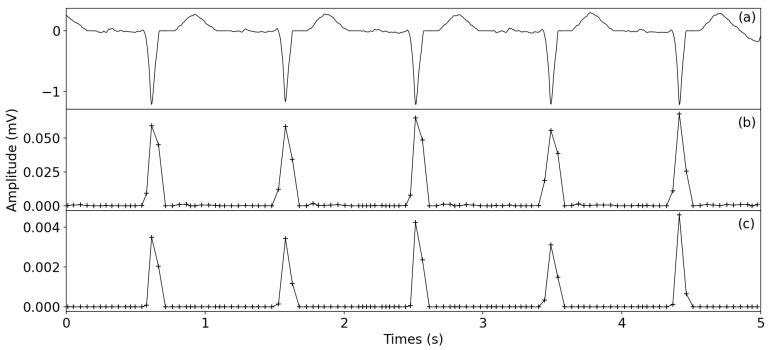
Comparison of original signal and the transformed signals, (**a**) Original signal of 207 from MIT-BIH database. (**b**) WVT—window variance transform signal. (**c**) SWVT—squared window variance transform signal.

**Figure 5 healthcare-09-00227-f005:**
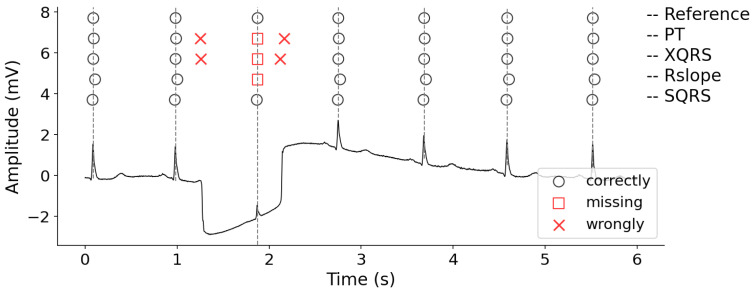
The R-peak detection results of 101 for different detectors on MIT-BIH database (a red X represents the wrongly identified R peak; a red rectangle is the missing R peak). TP—true positive (correctly identified R peak); FN—false negative (missing R peak); FP—false positive (wrongly identified R peak); SEN—sensitivity; PPV—positive predicative value; CSI—cumulative statistical index.

**Figure 6 healthcare-09-00227-f006:**
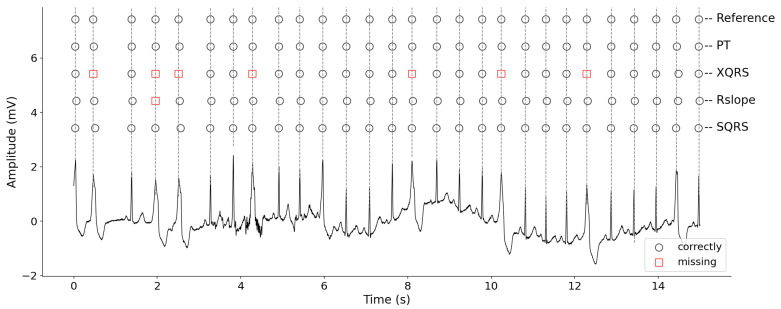
The R-peak detection results of 208 record for different detectors (a red rectangle is the missing R-peak). TP—true positive (correctly identified R peak); FN—false negative (missing R peak); FP—false positive (wrongly identified R peak); SEN—sensitivity; PPV—positive predicative value; CSI—cumulative statistical index.

**Figure 7 healthcare-09-00227-f007:**
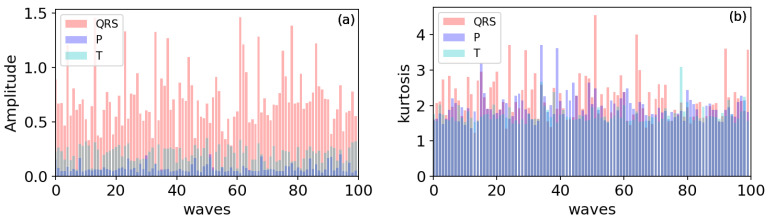
The comparison of QRS complexs distributions with P/T waves. All waves were extracted from the qtdb dataset: (**a**) amplitude; (**b**) kurtosis.

**Figure 8 healthcare-09-00227-f008:**
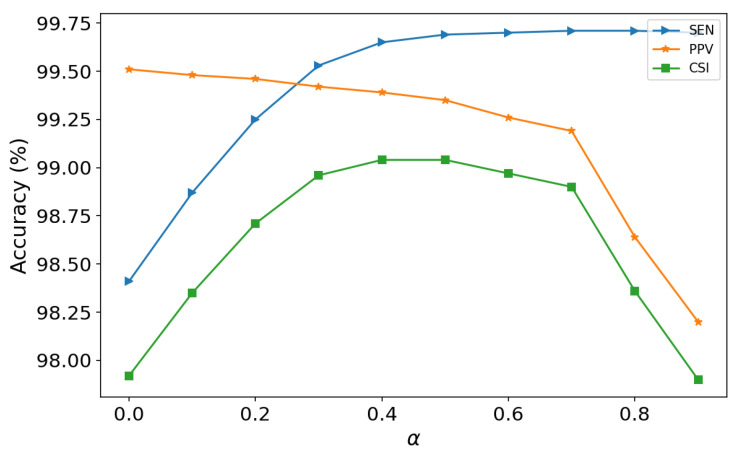
The performance with different α for SEN (sensitivity), PPV (positive predicative value) and CSI (cumulative statistical index). P wave—Indicates atrial depolarization, or contraction of the atrium; QRS complex—Indicates ventricular depolarization, or contraction of the ventricles; T wave—Indicates ventricular repolarization.

**Table 1 healthcare-09-00227-t001:** Results of evaluation of proposed method on MIT-BIH database.

Record	TP	FP	FN	SEN (%)	PPV (%)	CSI (%)
100	2273	1	0	100	99.96	99.96
101	1863	3	2	99.89	99.84	99.73
102	2174	13	13	99.41	99.41	98.81
103	2084	0	0	100	100	100
107	2128	5	9	99.58	99.77	99.34
112	2539	0	0	100	100	100
113	1795	0	0	100	100	100
114	1875	118	4	99.79	94.08	93.87
115	1953	0	0	100	100	100
116	2385	1	27	98.88	99.96	98.84
117	1535	0	0	100	100	100
118	2278	0	0	100	100	100
119	1987	1	0	100	99.95	99.95
121	1859	4	4	99.79	99.79	99.57
122	2475	1	1	99.96	99.96	99.92
123	1518	0	0	100	100	100
124	1617	2	2	99.88	99.88	99.75
200	2598	1	3	99.88	99.96	99.85
202	2114	18	22	98.97	99.16	98.13
205	2639	0	17	99.36	100	99.36
207	1842	133	18	99.03	93.27	92.30
208	2893	5	62	97.90	99.83	97.73
209	3005	0	0	100	100	100
210	2607	2	43	98.38	99.92	98.30
212	2748	0	0	100	100	100
213	3243	5	8	99.75	99.85	99.60
214	2247	37	15	99.34	98.38	97.72
215	3357	2	6	99.82	99.94	99.76
217	2190	22	18	99.18	99.01	98.19
219	2148	1	6	99.72	99.95	99.67
220	2048	0	0	100	100	100
221	2419	4	8	99.67	99.83	99.51
222	2463	0	20	99.19	100.00	99.19
228	2034	29	19	99.07	98.59	97.67
230	2256	0	0	100	100	100
231	1571	0	0	100	100	100
232	1779	69	1	99.94	96.27	96.21
233	3072	5	7	99.77	99.84	99.61
234	2753	0	0	100	100	100
Total	88,364	482	335	99.65	99.39	99.03

TP—true positive (correctly identified R peak); FN—false negative (missing R peak); FP—false positive (wrongly identified R peak); SEN—sensitivity; PPV—positive predicative value; CSI—cumulative statistical index.

**Table 2 healthcare-09-00227-t002:** Comparison of R-peak detection performance with some classic methods.

Method	SEN	PPV	CSI
PT	99.04	99.38	98.49
RSlope	99.23	98.58	97.83
XQRS	99.19	99.21	98.41
**SQRS**	**99.65**	**99.39**	**99.04**

Note: SEN = Sensitivity, PPV = positive predicative value, CSI = cumulative statistical index.

**Table 3 healthcare-09-00227-t003:** The detection error accuracy comparison results among the four methods.

Method	Total	FP	FN	$F_{d}$
PT	88,699	512 (0.58%)	754 (0.85%)	1.43%
RSlope	88,699	1198 (1.35%)	721 (0.81%)	2.16%
XQRS	88,699	715 (0.81%)	807 (0.91%)	1.72%
**SQRS**	88,699	**482 (0.54%)**	**335 (0.38%)**	**0.92%**

Note: FP = correctly identified R-peaks, FN = missing R-peaks, F_d_ = error rate.

**Table 4 healthcare-09-00227-t004:** The enhancement stages among the four methods.

Method	Preprocessing	Enhancement
PT	bandpass	Derivative Squaring Moving-window integration
XQRS	bandpass	Moving-window integration
RSlope	detrending lowpass	Separate heart beat detection Quality assessment
SQRS	moving-average	window variance transform

PT is a real-time QRS detection algorithm. XQRS is a QRS detection algorithm in WFDB application guide. Rslope is a QRS detection algorithm proposed by Gieraltowski et al. in 2015. SQRS is the proposed method.

**Table 5 healthcare-09-00227-t005:** The performance of the proposed method using a kurtosis threshold.

Using Kurtosis	TP	FP	FN	SEN	PPV	CSI
No	87,190	374	1509	98.44	99.52	97.96
Yes	88,364	482	335	99.65	99.39	99.04

TP is correctly identified R-peaks, FP is wrongly identified R-peaks and FN is missing R-peaks. SEN is sensitivity, PPV is positive predicative value and CSI is cumulative statistical index.

## Data Availability

All of the relevant data are presented within the manuscript. All data is public.

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
