# Peer review of "ECG Enhancement and R-Peak Detection Based on Window Variability"

_healthcare, 2021, doi:10.3390/healthcare9020227_

Round 1
Reviewer 1 Report
The authors propose a time-efficient method for R-peaks detection in ECG records. The paper is interesting and easy to read. The experimentation carried out is well described and shows that the proposed method outperforms existing solutions in the literature. In my opinion the paper deserves publication but after addressing some minor issues.
Although the paper is well written, I have some comments minor comments:
Lines 38-42 (For instance… And … R-peaks) The paragraph deserves to be rephrased.
Line 50-51. The method uses only one technique…: the authors should state why (do they consider it is enough? would the results be improved with the addition of other techniques?…)
Figure 2. Step 2 description: typo "squring" (I guess "squaring"…)
Line 80 and ss: the size of the window (w) is used but I see no reference to the procedure to fix its value. It seems to me that the same value is used to obtain both W_P and W_NP. A description of such procedure should be included.
Line 81: local peaks … is (ARE)
Line 100 and ss: Section 2.2.3 should be rewritten: Tc should be defined (line 101) and the way it is obtained (Equation (5)) substantiated; decision rules are referred to in the beginning of the section but described at the end; \alpha is used (with no reference to its role) and described as a “super parameter” in line 220 (Section 4.3)…
Reviewer 2 Report
-The paper presents a novel ECG enhancement and R-peaks detection method based on window variability.
-The topic and area of research are relevant.
-The technical aspects of the research presented seem sound and detailed.
-I suggest to be consistent when defining or indicating the meaning of acronyms by putting the first letter of each word in capital letters.
-In the first paragraph of the "Introduction" section "Figure 1" is referred to as "fig.1", which is not consistent to how the authors refer to figures and tables in the rest of the paper.
-I think the authors should present what are their plans for future work.
-In addition to the evaluation measures used, did the authors consider using other evaluation measures like AUC (Area Under the ROC Curve)?
-Have the users considered other datasets for their future work? It would be interesting to see if the preprocessing of raw data will have an effect when the proposed method is applied to data from a non-curated dataset.
